# Baby Food and Oral Health: Knowledge of the Existing Interaction

**DOI:** 10.3390/ijerph19105799

**Published:** 2022-05-10

**Authors:** Miriam Fioravanti, Gianni Di Giorgio, Roberta Amato, Maurizio Bossù, Valeria Luzzi, Gaetano Ierardo, Antonella Polimeni, Iole Vozza

**Affiliations:** Department of Oral and Maxillofacial Sciences, Sapienza University of Rome, 00185 Rome, Italy; gianni.digiorgio@uniroma1.it (G.D.G.); amato.1798983@studenti.uniroma1.it (R.A.); maurizio.bossu@uniroma1.it (M.B.); valeria.luzzi@uniroma1.it (V.L.); gaetano.ierardo@uniroma1.it (G.I.); antonella.polimeni@uniroma1.it (A.P.); iole.vozza@uniroma1.it (I.V.)

**Keywords:** oral health, diet, baby food, oral health and nutrition

## Abstract

Background: The purpose of this study is to verify parents’ knowledge of child nutrition and their awareness of the interaction between unhealthy sugars in their child’s diet and caries formation. Methods: a questionnaire was proposed using Instagram to analyze type of breastfeeding; type of weaning and diet; home oral hygiene maneuvers; bad habits (use of pacifiers, bottles, and sugary substances); knowledge on the usefulness of fluoride; and first dental visit. A total of 200 parents from different regions of Italy with children aged 2 months to 6 years were contacted. Results showed that 66% parents preferred breastfeeding, while the remaining 34% chose artificial breastfeeding. Fifty percent (100 babies) started weaning at six months, 20% (40 babies) at the fifth month, 13.5% (27 babies) at the fourth month, and only 11.5% (23 babies) in a range from the seventh to ninth month of life. Oral hygiene practices were performed only by 25% of parents before eruption of the first tooth. After eruption of the first tooth, there is greater attention to home oral hygiene practices: 59% of parents carry out and teach their children daily home oral hygiene maneuvers. Conclusions: it is possible to raise awareness among parents and caregivers on the importance of food education.

## 1. Introduction

Oral health in the growth phase is a goal to be pursued to improve the quality of life of the child as well as to ensure good aesthetics and correct social integration. Oral health is critical for important growth functions such as speaking, chewing, and swallowing [1].

A child’s oral health could be compromised by unhealthy lifestyles. The nutritional education of parents and, consequently, of children, is of primary importance to avoid the onset of developmental problems [2].

Parents should educate the child about proper nutrition and hygienic lifestyles by teaching the correct home oral hygiene habits from the first months.

Healthy nutrition is important at all ages, but even more so for children. A balanced diet, with an adequate intake of calories and nutrients and their correct distribution throughout the day, ensures optimal physical and cognitive development during the years of growth. Since a high sugar intake constantly accompanies children as they grow up, important nutrition education and coordination between families and institutions is mandatory [3]. 

A diet particularly rich in sugars promotes the formation of Early Childhood Caries (ECC), [4] loaded with deciduous teeth, severely damaging the child’s growth. The use of honey, sugary drinks, but also sugar added to milk (especially at night, when the salivary flow is very low) administered to the very young child through a pacifier or a bottle, determines the demineralization of the enamel of the deciduous teeth. These initial lesions can develop into carious processes, creating cavities and causing pain, compromising the quality of the baby’s sleep. Moreover, some eating patterns can influence not only daytime vigilance, but also night sleep. Behavioral problems and cognitive functioning are associated with sleep disruption in children. Disruption of sleep can reduce the health-related quality of life of children and adolescents and can worsen the severity of common gastrointestinal disorders [5]. 

If the caries is in an advanced stage, an extractive treatment should be considered. The early loss of a deciduous element can cause an alteration of the growth mechanisms of the oro-maxillofacial complex in terms of incorrect occlusion and incorrect muscle pattern, with impaired swallowing and phonation [6], and can cause suffering in psychological relationships in the patient pediatric [7]. The literature shows that dental caries in deciduous teeth has increased in recent years, and that, in permanent teeth, it has decreased [8]. Fundamental for the oral health of the child is the role of parents who, motivated, informed, and educated in dental prevention, implement oral health education interventions according to the child’s growth phases [9].

A diet is defined as healthy and balanced when it alternates daily all the nutrients in adequate quantities, ensuring the presence of the right balance between the intake of animal and vegetable proteins, simple and complex sugars (more bread, potatoes, pasta or rice and less sweets), animal fats, and therefore provides a fair amount of vitamins, minerals, and whole foods (important for their fiber content) [10].

Nutrition plays a vital role in the growth of the baby as proper nutrition plays an important role in the healthy growth and optimal development of the baby. From a regulatory point of view, the European Society of Gastroenterology, Hepatology and Pediatric Nutrition (ESPGHAN) pays great attention to baby food by the WHO and the Ministry of Health and again by the European Food Safety Authority (EFSA), from the American Academy of Pediatrics. In particular, attention is paid to the balance between the various nutritional components of foods (proteins, carbohydrates, fats, minerals, and vitamins), and, for some of them, there are minimum and/or maximum limits. It is essential that the parent learn to distinguish baby foods from those foods, which, due to their composition and safety, are not suitable for infants, even if marketed as such. The food par excellence is homogenized. The function of homogenization is to reduce the food into very fine particles that allow it to be taken without chewing, increasing digestibility. In reality, around the sixth month of life, the newborn’s intestine is now mature and able to digest all the nutrients introduced with food. Furthermore, if complementary feeding is offered in conjunction with the onset of neuromotor skills in the child, chewing is not only possible, but also desirable for the development of the orofacial muscles [11].

The aim of this study is to verify parents’ knowledge of their children’s oral health and the relationship between the food products used in the first years of life (during weaning) and oral health. Parental knowledge of food education and its influence on growth are therefore of primary importance. The hypothesis of the study is to verify the awareness of Italian parents on food education in relation to the oral health of their children.

## 2. Materials and Methods

The study protocol complies with the ethical guidelines of the 1975 Declaration of Helsinki. This survey was conducted from January to March 2020 by proposing a questionnaire, due to the SARS-COVID-19 pandemic, using a social network as an interactive online dialogue: Instagram^®^. This digital platform was chosen because it is more commonly used as a dialogue box than other social networks and has allowed us to gather more information. The inclusion criteria were to have male or female children aged 0–4, born within the last 7 years (2013–2020). Parents with children born before 2013 were excluded to prevent the reported data from being incorrect due to too much time. The online questionnaire was created and sent via the Google^®^ platform, and the participants signed an online informed consent to participate in the study.

This study made it possible to analyze the type of breastfeeding through questions; type of weaning and diet; home oral hygiene maneuvers; bad habits (use of pacifiers, bottles and sugary substances); knowledge on the usefulness of fluoride and the first dental visit around 6 years.

The questionnaire was carried out by a pediatric dentist and a community dentist; to collect the information of interest, a check list was created focusing on the variables considered most important by the literature in the pediatric nutrition field [12]. The questionnaire was already used in a pilot study through a degree thesis.

The data collected with the questionnaire were recorded with a specially designed computer program and collected in a Microsoft Excel database. Descriptive statistics were calculated for each item, and the percentage of participants who answered yes/no to each of the 16 items was calculated. Data analysis was performed using SPSS 25.0 for Windows (SPSS Inc., Chicago, IL, USA). The chi-squared test was performed to assess whether there was a significant difference in the frequencies with which the response alternatives were chosen. The results were considered significant at a *p*-value ≤ 0.05.

### Participants

A total of 200 parents with children aged between 2 months and 6 years, with an average of 18.82 months of age (standard deviation 12.45). Forty-six percent were male children, 54% female girls (Appendix A, Table A1).

The questionnaire consisted of open-ended questions (number 1, 4, 5, 11, 12, 13, 14, 15, 17) and closed-ended questions (number 2, 3, 6, 7, 8, 10, 16, 18, 19, 20, 21, 22, 23, 24, 25, 26, 27, 28, 29, 30).

The number of participants was established through a Power Analysis carried out with the G-Power program. The number of subjects required to obtain a power (1-β) of 0.95 and an effect size of 0.3 corresponding to a probability of error (α) of 0.05 was equal to 145.

## 3. Results

The online questionnaire was carried out with ease and commitment by all the parents involved in the study. Two hundred parents from different regions of Italy with children aged between 2 months and 6 years answered the questionnaire.

The results show that 66%, and therefore 132 parents, preferred breastfeeding, while the remaining 68 parents, or 34%, chose bottle feeding (χ^2^ = 20.48; *p* < 0.001; Appendix A, Figure A1).

Although the age range is quite large, 79.5% (about 159) children consume milk during their diet, and only 20.5% (41 children) do without it (χ^2^ = 69.62; *p <* 0.001; Appendix A, Figure A1). This means that the diet is based on the use of milk repeated several times a day. The daily frequency of intake varies.

Out of 200 responses, 50% (100 babies) started weaning at six months, 20% (40 babies) at the fifth month, 13.5% (27 babies) at the fourth month, and only 11.5% (23 babies) in the period between the seventh to the ninth month of life, 2,5% before 4 months (5 babies). No data are available for the remaining 2.5% of the children for whom the parents did not provide an answer (five babies).

There are two main methods of weaning:The classic: that is, it involves the use of products such as: homogenized, freeze-dried, prepackaged, or homemade, chosen by 75% of parents (150 parents);Self-weaning: which allows the feeding of the child to increase the solid food of the adult in small pieces, 25% (50 parents) was chosen in the questionnaire.

Another relevant figure is that 60.5% (121 parents) prefer not to use prepackaged foods, but rather to prepare baby food using homemade methods or with the aid of homogenizers. The remaining 39.5% is divided into two macro categories: the former is convinced of the safety of the packaged product and the latter use them due to lack of time.

Then, questions that are more specific were asked about the frequency of food consumption during the weekly diet. The first question analyzes which fruit purees are used and how often. Among the favorites are those with pear, mixed fruit, and all kinds of fruit.

In the weekly diet, fish and meat are also added 2–3 times a week according to the child’s needs (2 to 8 times a week). Vegetables are less consumed. The intake of cheeses is very varied: from zero times to 14 times a week (also considering parmesan on baby food). As for the frequency of use of ready meals, most mothers have used them with an average consumption of 3–4 times a week.

Oral hygiene practices were performed only by 25% of parents before the eruption of the first tooth (χ^2^ = 50.00; *p* < 0.001; Appendix A, Figure A1)

Regarding the methods of execution: 29 used a gauze moistened with water or physiological solution; the remaining 21 parents used silicone thimbles or gloves.

After the eruption of the first deciduous tooth, there is greater attention to home oral hygiene practices: 59% of parents carry out and teach their children daily home oral hygiene maneuvers (χ^2^ = 6.48; *p* < 0.05; Appendix A, Figure A1).

The use of pacifiers or baby bottles was never carried out in 16.5% (χ^2^ = 89.78; *p <* 0.001); the remainder fail to stop using it, with the consequence of a spoiled habit. Eight percent of parents (16) also add sweetener to the pacifier or bottle, especially honey. In addition, 11% (23 parents) assist with their children’s sleep during the night using a bottle containing chamomile or fennel tea, and 19% (39 parents) use milk.

As regards the knowledge on the use of fluorinated products, only in 31% of cases is fluoride toothpaste (or with hydroxyapatite) is applied; the remaining 69% do not use toothpaste (χ^2^ = 28.88; *p <* 0.001); rather, only a toothbrush and water, or the use of natural toothpastes commonly bought at the supermarket and used by the whole family, are used. In fact, only 5% of parents were able to specify the percentage of fluoride in toothpastes.

In fact, only 9.5% of parents administer fluoride to their offspring while the remaining 90.5% do not did (χ^2^ = 131.22; *p <* 0.001). Furthermore, only 22.5% of parents choose the type of water to consume in the family according to the percentages of fluoride present in it (χ^2^ = 60.50; *p <* 0.001; Appendix A, Figure A1).

As for the question relating to the importance of a first dental visit within 6 years of age, 88.5% of parents consider a first dental visit at six years of age to be fundamental and useful; 11.5% declared instead that they do not consider it necessary (χ^2^ = 118.58; *p* < 0.001; Appendix A, Figure A1).

## 4. Discussion

The growth of the child is the focus of numerous projects involving the scientific research of pediatricians, pediatric nurses, midwives, dentists and dental hygienists, family practice doctors, clinical officers, health assistants, and nutritionists working in the public and private sector [13]. Several studies have shown a correlation between baby nutrition, systemic health, and oral health. Feeding patterns adopted in childhood are closely associated with the risk of developing dental caries. As demonstrated by Athavale et al., focus groups have identified the junk food environment, busy family life, and limited dental care as contributors to ECC [14]. Breastfeeding is an unparalleled way of providing ideal food for the healthy growth and development of babies; it is also an integral part of the reproductive process with important implications for the health of mothers. Analysis of the evidence has shown that, on a demographic basis, exclusive breastfeeding for 6 months is the optimal way to feed babies.

Infants should then receive complementary foods while continuing to breastfeed up to 2 years of age or beyond. Breastfeeding is also recommended, because it significantly contributes to the thrust and growth of the oromaxillofacial district, as well as conferring immunological components in nourishing milk [14].

With growth, the nutritional needs of the infant increase, and it is therefore advisable to add complementary foods.

The transition from exclusive breastfeeding to family feeding, called complementary feeding, typically covers the period from 6 to 18–24 months of age and is a very vulnerable period. Using the online questionnaire, our study analyzed the methods of breastfeeding and the habits adopted by Italian parents. The answers to the questionnaire on weaning show that 50% of parents start around six months and the other half choose different periods. Some parents decide to postpone the posting after six months and still others before six.

The WHO does not indicate a precise age, because the factors that determine the appropriate time for weaning are linked to the level of development of each individual child. From this study, it emerges that Italian parents prefer pears, mixed fruit, and all types of fruit because they are tastier for the child, even if they are high in sugar.

The intake of fish and meat is very subjective, but the use of vegetables is quite low in almost all children. The intake of cheeses varies according to the tastes of the child but, in most cases, there is a high frequency of intake. The choice of snacks often exposes children to products that offer concentrated energy with low nutritional value.

The use of sugary drinks during the day is very common in children, as shown by data from the National Health and Nutrition Examination Survey (NHANES): in the period 2011–2014, 62.9% of young people consumed at least one sweetened drink a day. Overall, children between the ages of 2 and 19 consumed an average of 143 kcal from sugary drinks on any given day. Children between the ages of 2 and 5 consumed 65 kcal and sugary drinks, which contributed 4.1% to total daily calories. In other countries such as Australia, there is also an increase in the prevalence of childhood obesity. As early as the age of five, one in five Australian children suffer from overweight or obesity. Obesity prevention strategies must therefore begin before this age [15].

Comparing our results with those of Masztalerz-Kozubek et al.’s 2020 study [16], Italian parents use added sugars more in their child’s diet. In fact, about 8% of parents use honey as an added sweetener. These sugar levels are lower than those shown in the Elliott et al. 2015 study [16], which specified that 60% of children consume drinks with added sugar. This can lead to a high risk of obesity, the risk factors of which are excessive consumption of high-density energy, nutrient-poor foods, and a lack of physical activity.

From the results of our study, this figure seems to increase: the results show that 19.5% of parents use milk to put their baby to sleep. A better understanding of the regulation of food intake during pregnancy can provide a basis for determining strategies that prevent maternal malnutrition, better improve the health of the fetus, and reduce the economic burden on mothers and babies [16]. Knowledge and awareness of the parental figure plays the most important role; in fact, the bad habits of the adult are always reflected in those of the child. Inadequate or excessive intake of micronutrients during pregnancy can have a negative impact on maternal and child health [14].

Fortunately, accumulated evidence suggests that, starting before birth and continuing throughout development, there are repeated and varied opportunities for babies to learn to enjoy the flavors of healthy foods. Maternal factors influence bacterial acquisitions, while colonization is mediated by oral health behaviors and practices and eating habits [15]. Complementary nutrition should therefore be timely, i.e., start from 6 months onwards. It should be adequate, in terms of quantity, frequency, and consistency, and possess a variety of foods that are safely prepared and administered. In addition, it should be administered appropriately according to the age of the child [14]. Proper complementary nutrition requires the intake of all nutrients in the right proportions [17].

A diet characterized by excessive sugar intake could have negative influences on the growth of the child, with an increased risk of caries [18,19], cardiovascular disease (CVD) [20], type-2 diabetes mellitus (T2DM), metabolic syndrome, non-alcoholic fatty liver disease (NAFLD), SDB (sleep disordered breathing). Obesity, in turn, affects the duration and quality of sleep, with a decrease in insulin sensitivity, hyperglycemia, and prevalent cardiometabolic risk factors. Researchers have been looking for a link between long-term high sugar consumption and ADHD [21,22]. The American Heart Association (AHA), the American Academy of Pediatrics (AAP), and the World Health Organization (WHO) recommend limiting free sugar intake to less than 10% of the total adult energy intake for children, noting that a further 5% reduction would provide additional WHO health benefits [23]. In addition to diet, they have been recognized in the etiopathogenesis of carious diseases salivary flow (quality and quantity), the immune system, age, and socioeconomic status, level of education, lifestyle behaviors, oral hygiene, and use of fluorides [24].

It is widely demonstrated that there is a direct correlation between caries diseases and dietary sugar intake, both in terms of frequency and quantity. The feeding style adopted by the parents therefore determines the oral health of the child. In questions related to bad habits, such as improper use or long-term use of the pacifier or bottle, most parents said they use it to make the baby fall asleep. A prolonged bad habit can cause growth alterations of the oro-maxillofacial complex [25]. Moreover, the use of a pacifier or bottle is often associated with the addition of a sweetener, with a calming and sleep-inducing effect. In our study, only 14 parents reported adding a sweetener to a pacifier or bottle. Allowing the child to remain in contact with drinks other than water for the whole night causes a prolonged lowering of the salivary pH, thus creating a favorable environment for the formation of carious lesions.

A meta-analysis of cross-sectional studies showed that breastfed infants were less affected by dental caries than bottle-fed infants were. Four studies showed that bottle-fed babies had more caries (*p* < 0.05), while three studies found no such association (*p* > 0.05). Scientific evidence has therefore indicated that breastfeeding can protect against dental caries in early childhood. WHO/UNICEF guidelines [26] recommend the benefits of breastfeeding for up to two years. Several studies aim to investigate the associations between feeding frequency at the age of 12 months and the prevalence of caries at the age of 3.

In 2018, Feldens et al. evaluated the diet of 345 infants, consisting of all foods and beverages consumed at the age of 12 months, including bottle use and breastfeeding. They were recorded using two 24-h infant diet boosters with mothers. The prevalence of early childhood caries (ECC) and severe ECC (S-ECC) at the age of 38 months was compared in groups defined by feeding frequency at 12 months, using regression models to fit sociodemographics and total carbohydrate intake. The prevalence of ECC at 38 months was 1.8 times higher in infants who breastfeed more than three times a day, 1.4 times higher in bottle-fed infants who are fed more than three times a day, and 1.5 times higher in infants with a combined high frequency of breastfeeding and bottle feeding together. This means that high frequency feeding in late childhood, including bottle use and breastfeeding, has been positively associated with early childhood dental caries, suggesting possible early targets for caries prevention [27].

Our results agree with the study conducted by Devenish et al., 2020. An analysis was performed to assess the prevalence of ECC in Australian preschool children. Breastfeeding practices were reported at 3, 6, 12, and 24 months of age. Free sugar intake was assessed at 1 and 2 years of age. Although there was no independent association between breastfeeding older than 1 year and ECC, the only factors independently associated with ECC were high free sugar intake. For this reason, breastfeeding is always strongly promoted in line with global and national recommendations.

However, to reduce the prevalence of early childhood caries, more efforts are needed to limit foods rich in free sugars [28]. Therefore, the etiological factor of ECC is not milk intake, but rather the excessive amount of sugar present in baby food consumed at five or more meals throughout the day. Many products that are often marketed and consumed by infants and young children contain sugars in amounts other than that which are stated nutrition labels and often above the recommended daily levels.

Further support needs to be provided to add more comprehensive sugar labeling to food and beverage products, especially those marketed or commonly consumed by children. Sugar and sodium levels in packaged foods are important public health factors to evaluate; however, little is known about the sugar and salt contained in packaged foods intended for our younger consumers. Elliott et al. [16] analyzed sugar and sodium levels in foods following US and American Heart Association medical guidelines: 15% of baby foods exceed the recommended moderate sodium level and 45% of the products have high sugar levels. Moreover, 60% of children are used to taking food and drink with added sugars, jeopardizing the quality of the diet. The transition year, with the withdrawal of breast milk and infant formula, is a period in which nutritional needs are high and the quality of the diet is often precarious. Rapid growth, along with the brain and cognitive development, require high-quality nutrition.

In the third part of the questionnaire, the parents’ knowledge of the child’s oral hygiene was assessed. Unfortunately, not all parents know the aids (gauze, silicone thimbles or gloves) to perform home oral hygiene maneuvers.

For this, it is necessary to increase the prevention campaigns for the oral health of children and of adults, emphasizing that the systemic health of the child starts from the oral cavity. In Italy, through a survey of 101 parents, it emerged that 57% of parents have never brushed the teeth of preschool children before the age of 3, 30% use pacifiers, and 17% use milk with a bottle all night. Even today, parents are not fully trained and informed about their children’s oral hygiene management, and the need for a parental oral health promotion program to control the oral health risk status of preschool- and school-aged children [29].

Regular removal of the biofilm, preferably with a fluoride-containing toothpaste, delays or even stops the progression of the carious lesion. This can occur at any stage in the progression of the lesion because it is the biofilm on the surface of the tooth or the cavity that drives the caries process [30].

In the last part of the questionnaire, the use of fluoride was evaluated in the form of drops or tablets. The acquired data highlighted the lack of knowledge of many Italian families of the chemical-physical properties of fluoride in the oral cavity. Insufficient exposure to fluorinated compounds represents a risk factor for the onset of carious lesions [31].

To control the development of caries, it is therefore necessary to promote good habits through a healthy diet rich in nutrients and oral hygiene practices and to eliminate bad habits such as a cariogenic diet [12].

Regarding the need, according to the parents, to evaluate the oral health of their child at about six years of age through a first dental visit, out of 200 parents, 11.5% responded negatively, a percentage still too high. It is advisable to carry out the first dental visit around 18/24 months regardless of the presence or absence of dental problems [31]. Caries risk assessment is therefore complex and includes physical, biological, environmental, and behavioral factors. A high concentration of cariogenic bacteria, inadequate eating habits, inadequate salivary flow, insufficient exposure to fluoride, poor oral hygiene, and low socioeconomic status are recognized as important risk factors for the disease [32].

The individual risk of developing caries lesions must be assessed through the experience of caries, eating habits and oral hygiene, fluoroprophylaxis and the general health of each individual, as well as through the socioeconomic status of the family [31]. It is therefore necessary to promote prevention and health education campaigns on the correlation between oral health and infant nutrition, involving parents, nursery and nursery school workers, and pediatricians.

### Limitations of the Study

The sample should be larger to gain a global view of the correlation between oral and systemic health in children.

It is therefore possible to plan a first dental visit accompanied by a complete medical history and a questionnaire that allows for the evaluation of the diet of the young patient. It would thus be possible to evaluate the influence of the type of diet on the risk of developing oral carious lesions.

## 5. Conclusions

This study made it possible to obtain a global vision of the current situation regarding the knowledge of Italian parents in the field of nutrition education and its influence on oral health. From our results, we can state that, unfortunately, Italian parents’ knowledge of the correlation between oral health and systemic health is not yet sufficiently widespread.

This could create alterations in the psychosocial and physical development of the growing subject. The prevention and education campaigns for the oral health of children carried out by dentists and dental hygienists are therefore fundamental. Nutrition education and the correct approach to oral health should therefore begin already during pregnancy and continue as a path of care as the child grows.

## Data Availability

Data available upon request due to restrictions. The data presented in this study are available upon request. The data is not publicly available for privacy reasons.

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
