# Peer review of "Baby Food and Oral Health: Knowledge of the Existing Interaction"

_ijerph, 2022, doi:10.3390/ijerph19105799_

Round 1
Reviewer 1 Report
The study aimed to verify parents' knowledge of child nutrition and awareness of the interaction between unhealthy sugars in a child's diet and caries formation.
In the methodology, the authors state, "200 parents from different 15
regions of Italy "and" 200 parents with children aged between 2 months and 6 years ..). However, the same does not correspond to the results in Table 1, where there are only 100.
Statistical methods are listed in 2 places - lines 108-113 and 120-123. Way ??
How is the sample size calculated?
What is the reliability and validity of the questionnaire?
List the parts of the questionnaire with questions (open, closed questions)?
Table 1 is unclear?
What is the name and meaning of graph A1? Yes / no answers are not frequencies.
The conclusion is general and not related to this study.
References are not written according to the instructions.
Once again, the idea of the study is very interesting, but the realization has many, many shortcomings, especially the methodology.
Author Response
Response to Review Comments
Baby food and oral health: knowledge of the existing interaction
Miriam Fioravanti *, Gianni Di Giorgio, Roberta Amato, Maurizio Bossù, Valeria Luzzi, Gaetano Ierardo, Antonella Polimeni, Iole Vozza
Dear Editor,
We wish to submit an original research article entitled “Baby food and oral health: knowledge of the existing interaction” for consideration by IJERPH.
We confirm that this work is original and has not been published elsewhere, nor is it currently under consideration for publication elsewhere.
We believe that this manuscript is appropriate for publication by IJERPH because your journal offer an important space to the dental field.
We have no conflicts of interest to disclose.
Please address all correspondence concerning this manuscript to me at miriam.fioravanti@unioma1.it
Thank you for your consideration of this manuscript.
Sincerely,
Miriam Fioravanti
We are very thankful for the thorough review. The paper was revised in the light of the reviewers' useful suggestions and comments. We hope the revision could meet your approval.
Response to Reviewer #1
Comment 1: The study aimed to verify parents' knowledge of child nutrition and awareness of the interaction between unhealthy sugars in a child's diet and caries formation.
In the methodology, the authors state, "200 parents from different 15
regions of Italy "and" 200 parents with children aged between 2 months and 6 years ..). However, the same does not correspond to the results in Table 1, where there are only 100.
Response 1: we corrected the table with sociodemographic data and patient data analyzed in the study.
Comment 2: Statistical methods are listed in 2 places - lines 108-113 and 120-123. Way ??
Response 2: We modified material and methods and reported statistical analysis correctly.
Comment 3: How is the sample size calculated?
Response 3: […]The number of participants was established through a Power Analysis carried out with the G-Power program. The number of subjects required to obtain a power (1-β) of .95 and an effect size of 0.3 corresponding to a probability of error (α) of .05 was equal to 145.
Comment 4: What is the reliability and validity of the questionnaire?
Response 4: The questionnaire consists of a series of questions aimed to analyzing different items whose together could give us a global vision of the current situation regarding the parental knowledge of the child's oral health.
The reliability of the questionnaire was previously analyzed during the drafting of a degree thesis, which presented this questionnaire as an object of study.
Comment 5: List the parts of the questionnaire with questions (open, closed questions)?
Response 5: As reported in material and methods […The questionnaire consisted of open-ended questions (number 1, 4, 5, 11, 12, 13, 14, 15, 17) and closed-ended questions (number 2, 3, 6, 7, 8, 10, 16, 18, 19, 20, 21, 22, 23, 24, 25, 26, 27, 28, 29, 30)…]
Comment 6: Table 1 is unclear?
Response 6: we modified Table 1
Comment 7: What is the name and meaning of graph A1? Yes / no answers are not frequencies.
Response 7: We modified graph A1.
Comments 8: The conclusion is general and not related to this study.
Response 8: we modified the conclusion.
Comment 9: References are not written according to the instructions.
Once again, the idea of the study is very interesting, but the realization has many, many shortcomings, especially the methodology.
Response 9: we modified references according to the instruction.

Reviewer 2 Report
All my comments have been adressed.
Author Response
Response to Review Comments
Baby food and oral health: knowledge of the existing interaction
Miriam Fioravanti *, Gianni Di Giorgio, Roberta Amato, Maurizio Bossù, Valeria Luzzi, Gaetano Ierardo, Antonella Polimeni, Iole Vozza
Dear Editor,
We wish to submit an original research article entitled “Baby food and oral health: knowledge of the existing interaction” for consideration by IJERPH.
We confirm that this work is original and has not been published elsewhere, nor is it currently under consideration for publication elsewhere.
We believe that this manuscript is appropriate for publication by IJERPH because your journal offer an important space to the dental field.
We have no conflicts of interest to disclose.
Please address all correspondence concerning this manuscript to me at miriam.fioravanti@unioma1.it
Thank you for your consideration of this manuscript.
Sincerely,
Miriam Fioravanti
We are very thankful for the thorough review. The paper was revised in the light of the reviewers' useful suggestions and comments. We hope the revision could meet your approval.
Reviewer 3 Report
4. Discussion
Surprisingly, it is only ONE paragraph at the (L. 182-342); is it possible for the authors to create some paragraphs?
Figure A1:
Is it possible for the authors to remove the Figure A1 from the manuscript?
Author Response
Response to Review Comments
Baby food and oral health: knowledge of the existing interaction
Miriam Fioravanti *, Gianni Di Giorgio, Roberta Amato, Maurizio Bossù, Valeria Luzzi, Gaetano Ierardo, Antonella Polimeni, Iole Vozza
Dear Editor,
We wish to submit an original research article entitled “Baby food and oral health: knowledge of the existing interaction” for consideration by IJERPH.
We confirm that this work is original and has not been published elsewhere, nor is it currently under consideration for publication elsewhere.
We believe that this manuscript is appropriate for publication by IJERPH because your journal offer an important space to the dental field.
We have no conflicts of interest to disclose.
Please address all correspondence concerning this manuscript to me at miriam.fioravanti@unioma1.it
Thank you for your consideration of this manuscript.
Sincerely,
Miriam Fioravanti
We are very thankful for the thorough review. The paper was revised in the light of the reviewers' useful suggestions and comments. We hope the revision could meet your approval.
Response to Reviewer #3
Comment 1: 4. Discussion Surprisingly, it is only ONE paragraph a la (L. 182-342); Is it possible for authors to create paragraphs?
Response 1: we have created the requests.
Comment 2: Figure A1: Is it possible for authors to remove figure A1 from the manuscript?
Response 2: We removed Figure A1.

Round 2
Reviewer 1 Report
The work has improved compared to the initial state. However, the references do not comply with the journal's instructions. What is reference number 13 ???
Author Response
please see the attachment.
Kindly,
Miriam Fioravanti

This manuscript is a resubmission of an earlier submission. The following is a list of the peer review reports and author responses from that submission.
Round 1
Reviewer 1 Report
First of all, thank you for the opportunity to review this manuscript.
The purpose of this study is to verify parents' knowledge of a child's nutrition and awareness of the interaction between unhealthy sugars in a child's diet and caries formation.
The idea of the study is very interesting, but the realization has many shortcomings. Therefore, the following are suggestions for the present manuscript:
INTRODUCTION:
- Line 43 – 50 – references are missing
- Line 52 – 69 – references are also missing
- Please specify the hypothesis of the study.
METHODS:
The methodology is confusing and poorly written.
- Which were inclusive, and which are the excluding criteria for the respondents?
- Why only Instagram?
- How were the respondents selected?
- Why were not take demographic data of parents such as education, economic status, age ... all this data can affect knowledge.
- In what form was the questionnaire sent to respondents - google form or some other???
- Please explain this: "The questionnaire was prepared together with a statistician that established the sample size sufficient for study validity." First, the questionnaire must be prepared with a psychologist, not a statistician. What statisticians could even possibly know about this topic. Who validated the questionnaire on what sample size and how?
- What is the reliability and validity of the questionnaire? Please include Cronbach's α.
- List the parts of the questionnaire with questions (open, closed questions)?
- Furthermore, based on what the statistician concluded, the minimal sample size number, and what number is that? 200? The minimal sample size needs to be calculated based on the total number of parents who have children from 2 months to 6 years in Italy. How was the sample size calculated?
- “The data gathered with the questionnaire were recorded with a specially designed computer program and collated in a Microsoft Excel database.” What special program???
- “The variance analysis and chi-square test were used to investigate the relationship between the variables.” – please explain what statistical methods do you used? What variable?
- Participant section…. “46% were male children, 54% female girls, 89 (Appendix A, Table A1).” Were children or parents’ participants?
RESULTS:
- “The online questionnaire was carried out with ease and commitment by all the parents involved in the study.” – what does this mean? How do you know this?
- One short table and one figure are all results????
- The results are poorly presented; nothing is clear.
DISCUSSION:
- The results have not been sufficiently explained and discussed. As there is no hypothesis, we do not know whether it is accepted or not?
- Are there any more limiting factors? What is the strength of the study?
CONCLUSION:
- The conclusion is general, not related to this study.
REFERENCES:
- References are not written according to the journal instructions.
APPENDIX:
- I don't think the questionnaire is complete; the frequency of food intake is not offered.
Reviewer 2 Report
Introduction
The introduction should be improved, as it sometimes seems to be a list of sentences without a linking point. Line 49 The definition of a balanced diet should be improved because as it is defined it is incorrect.
Materials and Methods
The software used for the statistical analysis must be correctly defined since different versions are mentioned.
Results
Line 130 A paragraph has been repeated.
Statistical analysis should be improved, looking for relationships between variables, predicting the risk of early caries in the sample, etc...
In the graphs it is recommended to include the significance.
Discussion
Line 162 Many other professions are also involved in child development research.
Line 183, it is not clear from the data provided that the evidence is as indicated by the authors.
The limitations of the study should be further explored.
Reviewer 3 Report
The present research article is well written and designed. The methodologies were applied in a rigorous manner. I suggest to the authors to rewrite the first section of th discussion in a more clear way. I also suggest to add a graphical abstract in order to resume the paper.
Reviewer 4 Report
The manuscript describes the relationship between parents' knowledge of their children's oral health and the food products used in the first years of life (during weaning) in Italy (n = 100 children). The referee, in particular, shows the interest in the description of Discussion "Scientific evidence has therefore indicated that breastfeeding can protect against dental caries in early childhood. WHO / UNICEF guidelines recommend the benefits of breastfeeding for up to two years." (L. 186-188)
[Suggestions]
1. Introduction:
The authors need to describe the rationale / research questions (such as the reason why the authors need to investigate the relationship between parents' knowledge of their children's oral health and the food products used in the first years of life (during weaning) in Italy.)
3. Results, and Graph A1:
The referee feels that the authors need to modify/creat new Tables/Figures in order to make the readers easily understood.
4. Discussion, and References:
The authors need to check the accuracy of the cited References # (such as from [6] to [20]).
For example,
L. 187: WHO / UNICEF guidelines [6]?
L. 190: In 2018 Feldens CA. et al, and L. 202 [7]?
L. 203: the study conducted by Devenish G. et al 2020, and L. 211 [8]?
L. 221: Elliott CD et al [9]?
L. 249: the Masztalerz-Kozubek D. et al 2020 study [11]?
[Typographical errors?]
L. 197: The authors need to check the meaning of "1.4 times m higher in bottle-fed infants".